# Long-Term Morbidity and Mortality after First and Recurrent Cardiovascular Events in the ARTPER Cohort

**DOI:** 10.3390/jcm9124064

**Published:** 2020-12-16

**Authors:** Marina Escofet Peris, Maria Teresa Alzamora, Marta Valverde, Rosa Fores, Guillem Pera, Jose Miguel Baena-Díez, Pere Toran

**Affiliations:** 1Unitat de Suport a la Recerca Metropolitana Nord, Institut Universitari d’Investigació en Atenció Primària Jordi Gol (IDIAP Jordi Gol), 08303 Mataró, Spain; maiteal2007@gmail.com (M.T.A.); martavalverdeperis@gmail.com (M.V.); rfores.bnm.ics@gencat.cat (R.F.); gpera@idiapjgol.info (G.P.); ptoran.bnm.ics@gencat.cat (P.T.); 2Masnou Primary Healthcare Centre, Gerència d’Àmbit d’Atenció Primària Metropolitana Nord, Institut Català de la Salut, 08320 Barcelona, Spain; 3Riu Nord-Riu Sud Primary Healthcare Centre, Santa Coloma de Gramenet, Gerència d’Àmbit d’Atenció Primària Metropolitana Nord, Institut Català de la Salut, 08921 Barcelona, Spain; 4Department of Emergency Service, Andorran Health Care Service, Hospital Nuestra Señora de Meritxell, AD700 Escaldes-Engordany, Andorra; 5La Marina Primary Healthcare Centre, Gerència d’Àmbit d’Atenció Primària Barcelona Ciutat, Institut Català de la Salut, 08038 Barcelona, Spain; josemibaena@gmail.com; 6Unitat de Suport a la Recerca de l’Àmbit de l’Atenció Primària de Barcelona Ciutat, Institut Universitari d’Investigació en Atenció Primària Jordi Gol (IDIAP Jordi Gol), 08007 Barcelona, Spain

**Keywords:** transient ischemic attack (TIA), ischemic stroke, myocardial infarction (MI), recurrent stroke, recurrent MI, peripheral arterial disease, ankle-brachial index

## Abstract

Background: Cardiovascular events are a major cause of mortality and morbidity worldwide. The risk of recurrence after a first cardiovascular event has been documented in the international literature, although not as extensively in a Mediterranean population-based cohort with low cardiovascular risk. There is also ample, albeit contradictory, research on the recurrence of stroke and myocardial infarctions (MI) after a first event and the factors associated with such recurrence, including the role of pathological Ankle-Brachial Index (ABI). Methods: The Peripheral Arterial ARTPER study is aimed at deepening our knowledge of patient evolution after a first cardiovascular event in a Mediterranean population with low cardiovascular risk treated at a primary care centre. We study overall recurrence, cardiac and cerebral recurrence. We studied participants in the ARTPER prospective observational cohort, excluding patients without cardiovascular events or with unconfirmed events and patients who presented arterial calcification at baseline or who died. In total, we analyzed 520 people with at least one cardiovascular event, focusing on the presence and type of recurrence, the risk factors associated with recurrence and the behavior of the ankle-brachial index (ABI) as a predictor of risk. Results: Between 2006 and 2017, 46% of patients with a first cardiovascular event experienced a recurrence of some type; most recurrences fell within the same category as the first event. The risk of recurrence after an MI was greater than after a stroke. In our study, recurrence increased with age, the presence of peripheral arterial disease (PAD), diabetes and the use of antiplatelets. Diabetes mellitus was associated with all types of recurrence. Additionally, patients with an ABI < 0.9 presented more recurrences than those with an ABI ≥ 0.9. Conclusions: In short, following a cardiac event, recurrence usually takes the form of another cardiac event. However, after having a stroke, the chance of having another stroke or having a cardiac event is similar. Lastly, ABI < 0.9 may be considered a predictor of recurrence risk.

## 1. Background

According to the World Health Organization, ischemic heart disease and stroke are the leading causes of death worldwide, resulting in 15.2 million deaths in 2016. There were 410,611 deaths in Spain in 2016, with a gross mortality rate of 884.0 deaths per 100,000 inhabitants. Over half of these were due to the top three causes of death: cancer, heart disease, and cerebrovascular diseases [1].

Stroke is one of the main causes of mortality and disability in adults [2]. After experiencing cerebrovascular disease (stroke or transient ischemic attack), about one-half of surviving patients are at increased risk of recurrent stroke and other subsequent vascular events [3]. Similarly, patients who survive a myocardial infarction are at high risk of major cardiovascular events such as recurrent myocardial infarction, stroke, or mortality [4].

Although both ischemic stroke and ischemic heart disease are arteriosclerotic diseases, they differ in some risk factors. Hypercholesterolemia, diabetes mellitus and male sex are more strongly associated with myocardial infarction while hypertension is more strongly linked to stroke [5].

On the other hand, peripheral arterial disease (PAD) of the lower extremities is a highly prevalent cardiovascular disease, particularly in the asymptomatic form, and is easy to diagnose with a simple, non-invasive tool: Ankle Brachial Index (ABI). An ABI of < 0.9 is considered abnormal and has not only been associated with a diagnosis of PAD, but it has also been a marker of incident cardiovascular events and mortality in both symptomatic and asymptomatic forms of PAD [6,7,8]. It has been repeatedly associated with a three to six times greater risk of cardiovascular events and mortality [9].

Various national and international studies have shown the association between ABI < 0.9 and the occurrence of vascular events, such as those by Criqui, Meves, Alzamora and Fowkes et al. [6,7,8,9].

There is also mixed research on the recurrence of stroke and MI after a first event [10,11,12] and the factors associated with such recurrence, including the role of pathological ABI, albeit with contradictory results [13,14,15,16,17].

While there does exist some literature on this matter [15,18,19,20], most of the research conducted in our country is focused exclusively on stroke recurrence.

One of the challenges of public healthcare is to reduce incidence and recurrence of these diseases; identification of the risk factors involved should be a key objective in primary care in order to implement preventive measures.

For these reasons, the objective of this study is to evaluate the recurrence of cardiovascular events in a Mediterranean population-based cohort with low cardiovascular risk. Our study focuses on the risk of experiencing another cardiovascular event—either coronary disease myocardial infarction (MI or angina) or cerebrovascular disease (stroke or transient ischemic attack)—after having suffered a first event. In addition, we study the relationship between cardiovascular event recurrence and ABI < 0.9.

## 2. Materials and Methods

### 2.1. Study Population

The ARTPER study is an ongoing, prospective, observational population-based cohort of 3786 randomly selected patients [8,21]. It was launched from September 2006 to June 2008 in the Barcelona area. The selection process included patients over the age of 49 ascribed to 28 primary health care centers in the Barcelona area.

At the time of recruitment, we performed an ankle-brachial index (ABI) on all participants. The following cardiovascular events were recorded at the beginning of the study and during follow-up every 6 months:Cardiac (MI or angina)Cerebral (stroke or transient ischemic attack)Symptomatic abdominal aortic aneurysm (SAAA)Vascular surgery, cardiovascular morbidity (any of the previous four types)Vascular mortality (presence of vascular cause)Overall mortality (vascular or non-vascular)Morbimortality (any of the events)

Events were defined as prevalent (occurring before enrolment in the ARTPER study) or incident (occurring after enrolment). While all incident events were taken into consideration, only the last event of each type (cardiac/cerebral/other) was considered for prevalent events. This is because the older the prevalent event was, the more difficult it became to obtain information about it. Two or more events of the same category occurring within 30 days were considered a single event.

### 2.2. Follow-up and Endpoint Adjudication

The randomly selected patients were followed until 2016 via phone calls every six months, analysis of the SIDIAP (Information System for Primary Care Research) database and systematic reviews of primary-care and hospital records. All the clinical incident events were confirmed with a medical committee formed by members who perform routine clinical practice. Clinical incidents were classified as cardiac, cerebral, or other. Vascular surgery and mortality were included in one of the three groups depending on the location of the surgery or cause of death, respectively.

### 2.3. Statistical Analysis

Continuous variables are expressed as means and standard deviations, and categorical variables are expressed as frequencies and percentages. Cardiovascular event recurrence was analyzed alongside the presence of PAD (ABI < 0.9) using recurrence during follow-up (yes/no) as a dependent variable and PAD as an explanatory variable, adjusting for several potential confounders. All tests were bilateral with a significance of 0.05. Stata v16 was used to perform the analysis.

### 2.4. Ethics Approval and Consent to Participate

All patients have been informed of the study and signed informed consent. The ethical committee IDIAP Jordi Gol has approved the study. Code P16/014

## 3. Results

Of the 3786 subjects, 2616 were excluded for not having a cardiovascular event, 271 for unconfirmed cardiovascular events, 341 for death and 38 for arterial calcification at baseline, with a total of 520 subjects remaining in the study (Figure 1).

Of the 341 subjects with a fatal first event, the cause was nonvascular in 239, vascular in 47 (31 cardiac, 8 cerebral and 8 others) and unknown in 39.

Subjects were followed for an average of 8.6 years (1.7 months–10.3 years), adding up to 32,695 person-years.

Table 1 shows the general characteristics of the participants: the mean age of the overall population included in the study was 69 ± 8 years; 67% were men; 25% of men and 15% of women had ABI < 0.9; and, as for other risk factors, 66% had hypertension, 63% were smokers at some point in time, 60% had hypercholesterolemia, and 29% had diabetes mellitus.

### 3.1. Prevalent Events

Prevalent events were identified for 276 (53%) subjects: 179 (65%) were cardiac events, 95 (34%) were cerebral events, and 25 (9%) were other events. There was one subject who could have had an event in more than one category.

### 3.2. Incident Events

A total of 513 incident events occurred in 350 subjects (67%). Among these patients, 61% had cardiac events, 36% suffered cerebral events, 15% had other events, and 13% died.

The sum of these percentages is greater than 100% because some patients may have had an event in more than one category. Among the entire sample (520 patients), 170 patients (33%) suffered prevalent events but not incident ones, 244 (47%) experienced incident events but not prevalent ones, and 106 (20%) suffered both prevalent and incident events.

### 3.3. Recurrent Events

The 520 patients included in the study suffered a total of 812 events (mean = 1.6 events/patient), ranging from 1 to 10 events per patient. There were 184 patients (35%) who suffered 2 or more events (recurrence).

Table 2 shows the recurrence event analysis, including the last prevalent event of each type and all incident events.

### 3.4. Analysis of Cardiac Recurrence

In 51% of MI events, no recurrent event followed. The main type of recurrence after ischemic heart disease was a cardiac event (33%), followed by stroke (9%) and 6% died in follow-up.

### 3.5. Analysis of Cerebral Recurrence

There was no recurrence after 65% of cerebral events. The main type of recurrence after a stroke was another stroke (17%), followed by a MI (15%), and 8% died during follow-up.

### 3.6. Analysis of Recurrence: ABI

Table 3 shows the association between ABI > 0.9/ABI < 0.9 and event recurrence, at the subject level.

As can be observed in Table 3, subjects with ABI < 0.9 presented higher rates of recurrence of both (cardiac and cerebral) events as compared to subjects with ABI ≥ 0.9.

Table 4 shows the effect of ABI on event recurrence.

As can be observed in Table 4, ABI < 0.9 doubles the risk of having any recurrence. Both in the raw model and when adjusting for other risk factors, there is clearly a greater risk of recurrence in patients with ABI < 0.9.

In the univariate model, age, ABI < 0.9, hypertension, diabetes, and the use of certain medicines (antihypertensives, antidiabetics, and antiplatelets) increased the risk of recurrence.

In the multivariate model, ABI < 0.9, diabetes and antiplatelets almost double the risk of recurrence in both the raw model and the model adjusting for different risk factors.

## 4. Discussion

ARTPER is a prospective study aimed at deepening our knowledge of cardiovascular disease.

Until now, the ARTPER population-based cohort with low cardiovascular risk has been used to evaluate the incidence of vascular events, improvements made to the reclassification of cardiovascular risk scales following the addition of the ABI, as well as low, borderline, and normal ABI as a predictor of incidents events [21,22,23]. This study is focused on prevalent and incident cardiovascular outcomes and their relationship with ABI to evaluate the recurrence of cardiovascular events in a general population over the age of 49.

As far as we know, ARTPER is the first population study carried out in Spain to evaluate incidence and recurrence in a prospective cohort at 10 years follow-up. The cardiovascular events are extremely well documented and confirmed via telephone calls and review of clinical records by a multidisciplinary team.

Our main findings are as follows:(1)The overall recurrence of events was 35.3%.(2)Most events occurred in the same category as the first event.(3)An ABI < 0.9 was associated with a greater recurrence of events, almost doubling recurrence with respect to subjects with ABI ≥ 0.9 in the raw model and after adjusting for age, hypertension, diabetes, statins and antiplatelets.(4)The risk factors associated with recurrence were age, smoking, high blood pressure, diabetes, and antiplatelets.

Previous studies of ABI and recurrent events carried out in Spain were not population-based [14,18,20]; they reported stroke recurrence of 9% (Alvarez-Sabin) [18] and 32.1% (Purroy) [14] at 12 and 18 months follow-up. Serena et al. [20] found vascular events occurred in 13.8% of patients at 6–12 months follow-up after inclusion. The difference in results as compared to our study (stroke recurrence 17%) may be due to the fact that these studies are based on hospital populations, with older patients and shorter follow-up.

As compared to other international studies, we detected a greater difference between MI and stroke recurrence. In our study, the risk of cardiac recurrence (33%) was greater than stroke recurrence (17%). In contrast, the Suying Li et al. study [11] with older patients with a prior MI or stroke, found a smaller difference between recurrence types: the cumulative incidence of recurrence at six years was 14.3% for recurrent MI and 13.4% for recurrent stroke. This difference may be due to the fact that the population in this study was older, was more predominantly female, that follow-up was at six years, or that there was some comorbidity, such as carotid stenosis and endarterectomy. Moreover, we also included revascularization and angina as cardiac recurrence, and their population was American, which presents a different cardiovascular risk than the Mediterranean population.

Along these lines, Han Jing et al. analyzed stroke recurrence in a rural population and found rates similar to those of the ARTPER study: the recurrent stroke rate was 22.5% at five years [11]. As in other studies, diabetes was an independent risk factor [11,24]. The ARTPER study looked at urban patients who had easy access to the hospital and primary health center.

Furthermore, Gerber et al. conducted a study in Minnesota on the epidemiology of MI and detected a 27.5% rate of recurrence of [25]. While this is similar to the ARTPER study (33%), the population age range was 50 to 90 years old. Recurrent MI was associated with a greater mortality risk than incident MI [12]. These findings suggest it is necessary to reduce the high recurrence rate in our population (33%) to improve the prognosis of these patients.

In our study, we found that patients with ABI ≥ 0.9 were less likely to present cardiovascular events or death. An ABI of <0.9 nearly doubled the recurrence of cardiovascular events. These results are similar to Lia Alves-Cabratosa et al. (retrospective cohort study), which found that patients with low ABI plus diabetes showed greater mortality, MI, and ischemic stroke risk [26]. They detected an HR for all-cause mortality of 1.42 (1.25–1.63) in the group with low ABI, but they did not focus on recurrence. The population was similar to the ARTPER group since it used SIDIAP (database originated from clinical records, a “real world” assessment) of the Catalan primary-care system but used a different age range (35–85 years). Overall, ABI may be a factor to consider in follow-up of these patients.

In the ARTPER study, following a cardiac event, there was a 33% rate of recurrence for MI and 9% for stroke. According to the literature, MI recurrence is frequent: a national study by Abu-Assi et al. found a MI recurrence rate of 2.5% person-years in the low-risk group and 15.5% person-years in the high-risk group [27], although they studied a hospital population (CardioCHUS register) and not a primary-care population as we did. They suggest that individual classification (low or high risk) of long-term events would make it possible to use resources more efficiently and design chronic management programs [27]. In this line, we would suggest classifying patients with ABI < 0.9 as high risk for recurrence and, thus, in need of further intervention and follow-up.

Furthermore, Abtan et al. conducted a study on a cardiovascular population with previous MI and found that residual ischemic risk after a MI progressively increased over four years of follow-up, with the primary outcome being the composite of cardiovascular death, MI, or stroke. Consequently, the authors recommend placing greater importance on intensive secondary prevention [28]. In contrast to our study, they did not analyze whether the recurrences were cardiac or cerebral.

In the ARTPER study, following a stroke, 17% of recurrences took the form of another stroke and 15% were MI. Upon comparing our findings on all forms of recurrence after a stroke with international studies, we identified similar results in a systematic review conducted by Boulanger et al. after strokes or transient ischemic attacks. The authors found that recurrent stroke is a more common cause of death than MI after an ischemic stroke/transient ischemic attack (TIA) [29].

According to our study, after a stroke, while there is a greater long-term risk of suffering a recurrent stroke than a MI, it is not twice as high, as Dhamoon et al. found at five years. They determined that the risk of recurrent stroke was 18.3%, whereas that of MI was 8.6% [30]. The difference in findings may be due to the length of follow-up and the cardiovascular risk behavior in the US population.

Based on our findings, we might also suggest that after a stroke there is a greater risk of having a MI than there is of having a stroke after a MI.

We also found that diabetes mellitus is associated with a higher rate of recurrence. Similarly, in a multicenter prospective study in Austria [31].

It is worth mentioning that the classic cardiovascular risk factors (smoking, hypertension, cholesterol, and diabetes) are fundamental aspects to control in secondary prevention. Our study population was characterized as follows: the mean age was 69 ± 8 years, participants were predominantly male, the obesity rate was 41%, 25% of men and 15% of women had ABI < 0.9, 66% were hypertensive, 60% had dyslipidemia, 47% were smokers, 29% had diabetes, and 4% had atrial fibrillation. Therefore, the number of patients with diabetes, high blood pressure and who smoke is not trivial. Regarding medication, 65% took antihypertensives, 53% statins, 23% antidiabetics, and 55% antiplatelets.

The ARTPER study has several strong points. Firstly, this is a study of a prospective, observational, population-based cohort with a long follow-up period of 10 years. Secondly, all cardiovascular events are extremely well documented and confirmed via telephone calls and review of clinical records by a multidisciplinary team (family doctors, neurologists, and cardiologists) every six months during this follow-up period. Thirdly, all unconfirmed events were excluded.

However, the ARTPER study also presents limitations. The first is the reduced number of recurrences, which we would like to expand upon in another study using the SIDIAP population database. The second limitation is that we do not know the exact date of diagnosis of some of the risk factors presented by the study population. This could influence the risk of recurrence, predictably with recurrence being greater the longer the evolution of the risk factor is.

There might also be arguments against including prevalent events since they are more difficult to verify. This is why we only included the last prevalent event for all three categories (cardiac, cerebral, other). However, excluding prevalent events leads to similar results, but with fewer cases (data not shown). Our analysis was focused on the event level (recurrence of each event, regardless of the number of events each individual had) and the individual level (patient has a recurrence or not). Analysis of the number of events (Poisson regression) led to similar conclusions. As for the analysis of the amount of time until recurrence (Cox regression), the sample was not large enough to yield interpretable results (data not shown).

Thirdly, we did not study subjects younger than 49 years old, because the prevalence of ABI < 0.9, cardiovascular events and recurrence is less frequent in younger people. Fourthly, we excluded patients with ABI ≥ 1.4 (arterial calcification) from our analysis since the behaviour of these subjects is different from those with peripheral arterial disease (ABI < 0.9) and a toe-brachial index was not performed on subjects with arterial calcification. Lastly, although HRs decreases when adjusting for normal vs. low comparisons, most of them are still statistically significant (*p* < 0.05).

## 5. Conclusions

In conclusion, the results of this study show that the risk of recurrence after a first cardiovascular event usually takes the same form, although the risk of other forms of recurrence is not negligible.

In the ARTPER cohort, the type of recurrence after a cardiac event is usually another cardiac event. However, after having a stroke the chance of having another stroke or having a cardiac event is similar

On the other hand, subjects with ABI < 0.90 present a greater risk of cardiovascular recurrence than those with ABI ≥ 0.90.

## Figures and Tables

**Figure 1 jcm-09-04064-f001:**
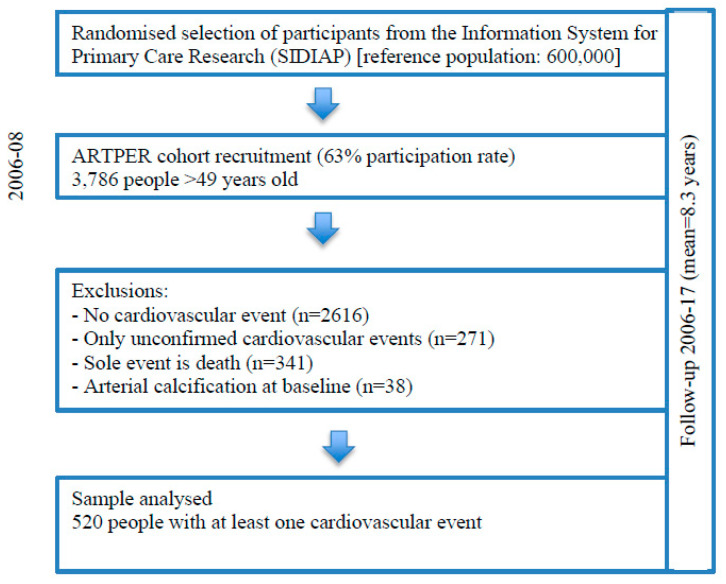
Flow chart of study participants.

**Table 1 jcm-09-04064-t001:** Baseline characteristics of the study participants (*n* = 520 patients).

	Total (*n* = 520)	
Age, years	69	±8
Female	173	(33%)
General obesity *	215	(41%)
Abdominal obesity **	317	(61%)
Ever smoker	327	(63%)
ABI *** < 0.9		
Men	88	(25%)
Women	26	(15%)
Comorbidity (medical records)		
Hypertension	341	(66%)
Hypercholesterolemia	311	(60%)
Diabetes	153	(29%)
Atrial Fibrillation	21	(4%)
Treatment/medication		
Antihypertensives	336	(65%)
Statins	274	(53%)
Antidiabetics	120	(23%)
Antiplatelets	287	(55%)
Subjects with prevalent events	276	(53%)
Cardiac	179	(34%)
Cerebral	95	(18%)
Others	25	(5%)
Subjects with incident events	350	(67%)
Cardiac	213	(41%)
Cerebral	126	(24%)
Others ****	51	(10%)
Death	44	(8%)

Results shown as mean ± standard deviation frequency (%), * Body mass index > 30 kg/m^2^, ** Waist circumference 102 cm (men) and 88 cm (women), *** ABI: Ankle Brachial Index, Others ****: Abdominal aortic aneurysms and other non-cardiac or cerebral vascular interventions.

**Table 2 jcm-09-04064-t002:** Recurrence of cardiovascular events (based on 520 patients and 812 events).

	Next Event									
Event	None *		Cardiac		Cerebral		Other ***		Death **	
Cardiac	190	51%	122	33%	34	9%	26	7%	21	6%
Cerebral	125	65%	29	15%	32	17%	7	4%	16	8%
Other ***	21	33%	11	17%	9	14%	22	35%	7	11%

* None refers to patients, who had just one event, ** Deaths are included in their respective cause (cardiac/cerebral/other), *** Other: Abdominal aortic aneurysms and other non-cardiac or cerebral vascular interventions.

**Table 3 jcm-09-04064-t003:** Association between ABI and event recurrence (based on 406/114 patients and 601/211 events (ABI ≥0.9/ABI < 0.9))

**ABI ≥ 0.9**										
**Next Event**										
**Event**	**None ***		**Cardiac**		**Cerebral**		**Other *****		**Death ****	
Cardiac	159	56%	88	31%	22	8%	15	5%	12	4%
Cerebral	106	71%	16	11%	23	15%	4	3%	11	7%
Other ***	12	31%	8	21%	5	13%	14	36%	6	15%
**ABI < 0.9**										
**Next Event**										
**Event**	**None ***		**Cardiac**		**Cerebral**		**Other *****		**Death ****	
Cardiac	31	35%	34	39%	12	14%	11	13%	9	10%
Cerebral	19	43%	13	30%	9	20%	3	7%	5	11%
Other ***	9	38%	3	13%	4	17%	8	33%	1	4%

* None refers to patients who had just one event, ** Deaths are included in their respective cause (cardiac/cerebral/other), *** Other: Abdominal aortic aneurysms and other non-cardiac or cerebral vascular interventions.

**Table 4 jcm-09-04064-t004:** Relationship between ABI and event recurrence (*n* = 520 subjects, 406 with ABI ≥ 0.9 and 104 with ABI < 0.9).

Outcome = Any Recurrence, Results for ABI < 0.9	OR	95%CI	*p*
Raw model	2.0	1.3–3.1	0.001
Adjusted for age	1.9	1.2–2.9	0.004
Adjusted for age, hypertension, and diabetes	1.7	1.1–2.7	0.014
Adjusted for age, statin and antiplatelet	1.8	1.2–2.8	0.009

Based on the analysis of risk factors associated with recurrence, it becomes clear that diabetes and age are significant risk factors (Table 5 and Table 6). Conversely, hypertension is not a significant risk factor for recurrence.

**Table 5 jcm-09-04064-t005:** Event recurrence and associated risk factors (*n* = 520 patients).

	Recurrence				
	No (*n* = 336)		Yes (*n* = 184)		
	*n*	%	*n*	%	*p*
Age, years	68	±8	70	±8	0.020
Gender					0.310
Male	219	(65%)	128	(70%)	
Female	117	(35%)	56	(30%)	
General obesity *					0.741
No	198	(59%)	106	(58%)	
Yes	137	(41%)	78	(42%)	
Abdominal obesity **					0.545
No	126	(38%)	74	(40%)	
Yes	208	(62%)	109	(60%)	
Ever smoker					0.893
No	124	(37%)	69	(38%)	
Yes	212	(63%)	115	(63%)	
ABI					0.001
≥0.9	277	(82%)	129	(70%)	
<0.9	59	(18%)	55	(30%)	
Hypertension					0.029
No	127	(38%)	52	(28%)	
Yes	209	(62%)	132	(72%)	
Hypercholesterolemia					0.193
No	142	(42%)	67	(36%)	
Yes	194	(58%)	117	(64%)	
Diabetes					0.001
No	253	(75%)	114	(62%)	
Yes	83	(25%)	70	(38%)	
Antihypertensives					0.007
No	133	(40%)	51	(28%)	
Yes	203	(60%)	133	(72%)	
Statin					0.267
No	165	(49%)	81	(44%)	
Yes	171	(51%)	103	(56%)	
Antidiabetics					0.002
No	273	(81%)	127	(69%)	
Yes	63	(19%)	57	(31%)	
Antiplatelets					0.002
No	167	(50%)	66	(36%)	
Yes	169	(50%)	118	(64%)	

Results shown as mean ± standard deviation or frequency (%), * Body mass index ≥ 30 kg/m^2^, ** Waist circumference ≥ 102 cm (men) and ≥ 88cm (women).

**Table 6 jcm-09-04064-t006:** Logistic regression for cardiovascular event recurrence (*n* = 520 patients)

	OR	95%CI	*p*	OR
ABI < 0.9	1.7	1.1	2.6	0.022
Age (by year)	1.0	1.0	1.0	0.053
DM *	1.8	1.2	2.7	0.004
Antiplatelets	1.7	1.1	2.4	0.009
**Outcome = Recurrence, Results for ABI < 0.9**				
Raw model	2.0	1.3	3.1	0.001
Adjusted for age	1.9	1.2	2.9	0.004
Adjusted for age, hypertension, and diabetes	1.7	1.1	2.7	0.014
Adjusted for age, and antiplatelet	1.8	1.2	2.8	0.009

* DM = Diabetes mellitus.

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
