# Peer review of "Long-Term Morbidity and Mortality after First and Recurrent Cardiovascular Events in the ARTPER Cohort"

_jcm, 2020, doi:10.3390/jcm9124064_

Round 1

Reviewer 1 Report

No additional comments.

Author Response

English language and style has been checked

No additional comments have been required. 

Reviewer 2 Report

Here are my comments:

  1. Methods page 3, line 93-100: The authors abruptly make this list without any context or explanation. What does it mean?
  2. How did the authors define “cardiovascular event”?
  3. How did the authors obtain data on arterial calcification in subjects in ARTPER cohort?
  4. Table 1 shows that only 55% of patients with prior cardiovascular events were on an antiplatelet and only 53% were on a statin. Why such low rates of treatment?
  5. Table 1, subjects with prevalent/incident events: What does “Others” mean? Please elaborate in detail.
  6. If the authors only included 520 subjects with at least one prior cardiovascular event, why were there prevalent events in only 276 (53%) of the subjects?
  7. For a supposedly “low risk” Mediterranean cohort, an incident event rate of 47% over 8 years seems high.
  8. Results, page 7, line 178: The text mentions that the risk was adjusted for: age, hypertension, diabetes, statins or antiplatelets; however, in Table 4 it states adjusted for age, statin and antiplatelet only. Why this discrepancy?
  9. Clinically, increased risk of recurrence with the use of antiplatelets (table 6a) does not make sense. How do the authors explain this finding?
  10. Ideally, table 5 should be table 2 – immediately following table 1, since it is a descriptive table, albeit in two groups (recurrence or no recurrence).
  11. Did all patients in the ARTPER cohort have an ABI?
  12. Discussion, line 223: “Treatment with antihypertensives, statins, and antiplatelet drugs reduced the risk of recurrence”. This statement is not supported by the author’s results which show higher risk of recurrence with antiplatelets.
  13. Discussion is exceedingly long and needs to be made more concise.
  14. ABI <0.90 simply means prevalent peripheral vascular disease which may be asymptomatic. Clearly, patients with PAD will have higher risk of cardiovascular events. Did the authors make any attempt to classify symptomatic vs. asymptomatic PAD in those with ABI<0.90?
  15. Based on what data do the authors claim that after having a stroke, the chance of another stroke or cardiac event is similar?

Author Response

Response to Reviewer 2

  • Methods page 3, line 93-100: The authors abruptly make this list without any context or explanation. What does it mean?

We performed an ABI on all participants, both at the time of recruitment and periodically times during the follow-up (every 6 months). Vascular events were countedon both.

We have modified the lines 92 to 100 for a better understanding of the cardiovascular events studied.

  • How did the authors define “cardiovascular event”?

From lines 92 to101 we define vascular events.

  • How did the authors obtain data on arterial calcification in subjects in ARTPER cohort?

 ABI was performed at the beginning of the study as baseline. Arterial calcification was considered when ABI was ≥1.4

  • Table 1 show that only 55% of patients with prior cardiovascular events were on an antiplatelet and only 53% were on a statin. Why such low rates of treatment?

Many studies have shown that the control of risk factors in secondary prevention is insufficient. The ARTPER study is a population-based study of real clinical practice and we have not performed any intervention in this regard. We believe there is an underprescription of these drugs.

Antiplatelet drugs or oral anticoagulants were used by 87.2%, antihypertensive medication by 84.4% and statins by 56.8% of stroke patients. 

Heuschmann PU, Kircher J, Nowe T, Dittrich R, Reiner Z, Cifkova R, Malojcic B, Mayer O, Bruthans J, Wloch-Kopec D, Prugger C, Heidrich J, Keil U. Control of main risk factors after ischaemic stroke across Europe: data from the stroke-specific module of the EUROASPIRE III survey. Eur J Prev Cardiol. 2015 Oct;22(10):1354-62. doi: 10.1177/2047487314546825. Epub 2014 Aug 19. PMID: 25139770.

Among the patients interviewed, the use of prophylactic drug therapies on admission, at discharge and at interview was as follows: aspirin or other antiplatelets drugs 47%, 90% and 86%; beta-blockers 44%, 66% and 63%; ACE inhibitors 24%, 38% and 38%; and lipid-lowering drugs 26%, 43% and 61%, respectively.

EUROASPIRE II Study Group. Lifestyle and risk factor management and use of drug therapies in coronary patients from 15 countries; principal results from EUROASPIRE II Euro Heart Survey Programme. Eur Heart J. 2001 Apr;22 (7):554-72. doi: 10.1053/euhj.2001.2610. PMID: 11259143.

The antiplatelet therapy rate in these two groups was 44.1% vs 86.5% (p < 0.005), the anticoagulant therapy rate was 11.4% vs 12.6% (no significant difference) and statin therapy rate was 65.2% vs 65.6% (no significant difference). 

Suárez C, Cairols M, Castillo J, Esmatjes E, Sala J, Llobet X, Palma JC; en representación de los investigadores del registro REACH España. Control de factores de riesgo y tratamiento de la aterotrombosis. Registro REACH España [Risk factor control and treatment of atherothrombosis. Spain REACH Registry]. Med Clin (Barc). 2007 Oct 6;129(12):446-50. Spanish. doi: 10.1016/s0025-7753(07)72882-8. Erratum in: Med Clin (Barc). 2008 Jan 26;130(2):46. PMID: 17953908.

  • Table 1, subjects with prevalent/incident events: What does “Others” mean? Please elaborate in detail.

Thanks very much for your comment, a foot table has been added in table 1 following the suggestions of the reviewer.

  • If the authors only included 520 subjects with at least one prior cardiovascular event, why were there prevalent events in only 276 (53%) of the subjects?

“Prevalent events” refers, in this study, to those events suffered before the study recruitment. In contrast, “Incident events” are those occurred after recruitment. This was explained in lines 102-106:

Events were defined as prevalent (occurring before enrolment in the ARTPER study) or incident (occurring after enrolment). While all incident events were taken into consideration, only the last event of each type (cardiac/cerebral/other) was considered for prevalent events. This is because the older the prevalent event was, the more difficult it became to obtain information about it. Twoor more events of the same category occurring within 30 days were considered a single event.

So 276 subjects had events before the start of the study, some of them having two events after. The others (520-276) had only one (or more) events exclusively after the study start.

  • For a supposedly “low risk” Mediterranean cohort, an incident event rate of 47% over 8 years seems high.

This 47% (=244/520) is over patients that already had a previous cardiovascular event, not a population risk since all patients with no events have been excluded.

  • Results, page 7, line 178: The text mentions that the risk was adjusted for: age, hypertension, diabetes, statins or antiplatelets; however, in Table 4 it states adjusted for age, statin and antiplatelet only. Why this discrepancy?

The correct version is the table. Text has been modified accordingly in line 181.

  • Clinically, increased risk of recurrence with the use of antiplatelets (table 6a) does not make sense. How do the authors explain this finding?

In our work, the different therapeutic groups or the doses of drugs used were not evaluated. This could partly explain the disagreement in antiplatelet use and the increased risk of recurrence due to the fact that we do not know if there may be an inappropriate prescription, underuse of combination therapies, or poor adherence to treatment.

Some secondary prevention studies in our country do not obtain a reduction in cardiovascular morbidity and mortality even if they are adequately treated in the intervention groups

Muñoz MA, Vila J, Cabañero M, Rebato C, Subirana I, Sala J, Marrugat J; ICAR (Intervención en la Comunidad de Alto Riesgo cardiovascular) investigators. Efficacy of an intensive prevention program in coronary patients in primary care, a randomised clinical trial. Int J Cardiol. 2007 Jun 12;118(3):312-20. doi: 10.1016/j.ijcard.2006.07.015. Epub 2007 Jan 29. PMID: 17261336.

Brotons C, Soriano N, Moral I, Rodrigo MP, Kloppe P, Rodríguez AI, González ML, Ariño D, Orozco D, Buitrago F, Pepió JM, Borrás I; PREseAP study research team. Randomized clinical trial to assess the efficacy of a comprehensive programme of secondary prevention of cardiovascular disease in general practice: the PREseAP study. Rev Esp Cardiol. 2011 Jan; 64(1):13-20. doi: 10.1016/j.recesp.2010.07.005. Epub 2010 Dec 30. Erratum in: Rev Esp Cardiol. 2011 Jun; 64(6):544. PMID: 21194823.

We agree with the reviewer, but we have not done more studies to investigate this result. This study could be carried out in the future

  • Ideally, table 5 should be table 2 – immediately following table 1, since it is a descriptive table, albeit in two groups (recurrence or no recurrence).

We had previously considered this suggestion from the reviewer, but finally we decided to put it as Table 5 because we are analyzing the recurrence of events according to the associated risk factors.

In any case, if the reviewer considers it essential, it could be changed.

  • Did all patients in the ARTPER cohort have an ABI?

Yes, they have at least one baseline ABI. Patients with no ABI (or not valid) were excluded from our cohort. This information has been added now to the Methods section (line 92).

  • Discussion, line 223: “Treatment with antihypertensives, statins, and antiplatelet drugs reduced the risk of recurrence”. This statement is not supported by the author’s results which show higher risk of recurrence with antiplatelets.

We have deleted point 5 (230 current line) since they did not correspond to the results obtained.

  • Discussion is exceedingly long and needs to be made more concise

We have eliminated some paragraphs to shorten the text as indicated by the reviewer.

We think that shortening the discussion could influence the understanding of the article.

  • ABI <0.90 simply means prevalent peripheral vascular disease which may be asymptomatic. Clearly, patients with PAD will have higher risk of cardiovascular events. Did the authors make any attempt to classify symptomatic vs. asymptomatic PAD in those with ABI<0.90?

Yes, it was studied. We have not separated the results into symptomatic and asymptomatic due to the size of the sample.

“Intermittent claudication was present in 10.0% of the subjects.” As the reviewer can see in this article: Alzamora MT, Forés R, Baena-Díez JM, Pera G, Toran P, Sorribes M, Vicheto M, Reina MD, Sancho A, Albaladejo C, Llussà J. The peripheral arterial disease study (PERART/ARTPER): prevalence and risk factors in the general population. BMC Public Health. 2010 Jan 27;10:38. doi: 10.1186/1471-2458-10-38. PubMed PMID: 20529387; PubMed Central PMCID: PMC2835682.

  • Based on what data do the authors claim that after having a stroke, the chance of another stroke or cardiac event is similar?

In table 2 the reviewer can see that after a stroke, next event is another stroke (17%) or cardiac event (15%). 

This manuscript is a resubmission of an earlier submission. The following is a list of the peer review reports and author responses from that submission.

Round 1

Reviewer 1 Report

The submitted manuscript is an observational study investigating recurrents events in 520 patients with at least one prior cardiovascular event.

Detailed comments.

Background. The introduction section is relatively unfocused. I suggest being more focused to clarify the background and main objective of the research.

Data presentation is somewhat confusing. The authors report the number of patients with prevalent events and incident events. However, the distinction is not clear, and all these numbers are rather confusing. I suggest to reformulate data presentation.

"However, the association between ABI and the recurrence of events has not received such attention." This sentence seems to express the main research question: whether or not ABI can predict recurrent events. However, this sentence does not hold true. A number of study have already evaluated the impact of ABI on event recurrences, for example:

- Stroke. 2019 Apr;50(4):853-858. doi: 10.1161/STROKEAHA.118.022180.
- Stroke. 2009; 40:3472–3477. doi: 10.1161/STROKEAHA.109.559278
- Stroke. 2016 Feb;47(2):317-22. doi: 10.1161/STROKEAHA.115.011321.

Therefore, the study results do not appear original. I suggest the authors to reconsider their main study question.

"However, there is a greater risk of having a cardiac event after a cardiac event than of having a stroke after a stroke." Again, the research seems not original as this concept has been clearly demonstrated in numerous previous reports.

Author Response

Response to Reviewer 1

  1. The introduction section is relatively unfocused. I suggest being more focused to clarify the background and main objective of the research

L57-85

Following the recommendation of the reviewer, we have reviewed and modified the introduction for a better understanding.

  1. Data presentation is somewhat confusing. The authors report the number of patients with prevalent events and incident events. However, the distinction is not clear, and all these numbers are rather confusing. I suggest to reformulate data presentation

L145-154

Following the suggestions of the reviewer, data presentation has been modified in the text (Results) and tables. Although the results and conclusions are the same, we expect that it will be easier to read and understand.

  1. "However, the association between ABI and the recurrence of events has not received such attention." This sentence seems to express the main research question: whether or not ABI can predict recurrent events. However, this sentence does not hold true. A number of studies have already evaluated the impact of ABI on event recurrences, for example:

- Stroke. 2019 Apr;50(4):853-858. doi: 10.1161/STROKEAHA.118.022180.
- Stroke. 2009; 40:3472–3477. doi: 10.1161/STROKEAHA.109.559278
- Stroke. 2016 Feb;47(2):317-22. doi: 10.1161/STROKEAHA.115.011321

L57-85

As we have commented in point 1, we have reviewed and modified the introduction for a better understanding. This sentence has been removed: "However, the association between ABI and the recurrence of events has not received such attention."

Therefore, the study results do not appear original. I suggest the authors to reconsider their main study question.

We agree with the reviewer that some of our findings are not new, specially the relationship between the types of first cardiovascular event with the recurrent one. However, ARTPER is an ongoing population-based cohort with more than 10 years of follow-up and almost 4,000 participants enrolled. We are in contact with our participants every 6 months, the accuracy of events are checked by a medical staff, there is a focus on peripheral arterial disease (PAD), and the context is the Mediterranean region*. Several interesting results from our cohort have been published by now, most of them included in high impact systematic reviews or meta-analyses. We think that it is interesting to learn how recurrence works in our cohort and how it relates to PAD to help better identify PAD and its potential effects and provide a better characterization of our cohort. In any case, we have added a sentence recognizing theshortcoming the reviewer points out and we have added some references provided by him/her.

*Please note the French paradox: the paradoxical situation in France with a low death rate from coronary heart disease despite the high consumption of saturated fat in the diet. The French paradox was later extended to other Mediterranean countries such as Spain, Italy, parts of the former Yugoslavia and Greece, where there were differences in both cardiovascular mortality and the incidence of myocardial infarction (MI) with respect to northern European countries. 

Dégano IR, Elosua R, Kaski JC, Fernández-Bergés DJ, Grau M, Marrugat J. Plaque stability and the southern European paradox. Rev Esp Cardiol (Engl Ed). 2013 Jan;66(1):56-62. English, Spanish. doi: 10.1016/j.recesp.2012.07.014. Epub 2012 Oct 15. PMID: 23078876.

  1. "However, there is a greater risk of having a cardiac event after a cardiac event than of having a stroke after a stroke." Again, the research seems not original as this concept has been clearly demonstrated in numerous previous reports

Please see our answer to the previous point.

Reviewer 2 Report

This is an interesting, well justified and valuable study.

The discussion requires revision to interpret the results and synthesise these interpretations with other published data .

There is no consideration of haemorrhagic versus ischaemic (vascular) stroke.

There is no discussion of the significant findings with respect to anti-diabetic and anti-hypertensive medications (whilst they are likely associated with the increased risk of diabetes and hypertension, this should be discussed).

There is no discussion of the significant finding with respect to ant-platelt drugs

L200-217 (and other parts of the discussion) merely repeats the results. Each point is repeated again later in the discussion.

L214 and 268 and 47 (abstract). the claim that after a stroke there is greater risk of suffering a recurrent stroke than an MI is not supported by the results (17% n=32 stroke vs 15% n=29 MI).

L273-274 Similarly - no statistics are presented to support the statement.

Much of the discussion merely provides detail of other related studies and reports their outcomes compared to the current study and does not actulayy discuss or interpret.

Author Response

Response to Reviewer 2

  1. The discussion requires revision to interpret the results and synthesise these interpretations with other published data.

L207-233

L220-233

We have reviewed the discussion section.

  1. There is no consideration of haemorrhagic versus ischaemic (vascular) stroke.

Indeed, there is no consideration of haemorrhagic versus ischaemic (vascular) stroke. The ARTPER cohort, from which the subjects in this study come, was designed to collect ischemic vascular events; haemorrhagic strokes were not recorded.

  1. There is no discussion of the significant findings with respect to anti-diabetic and anti-hypertensive medications (whilst they are likely associated with the increased risk of diabetes and hypertension, this should be discussed).

While we fully agree with your comments, in the present study the drugs were only analysed at the time of inclusion in the study and they were not prospectively followed up.

  1. L200-217 (and other parts of the discussion) merely repeats the results. Each point is repeated again later in the discussion.

We have removed these lines.

  1. L214 and 268 and 47 (abstract). the claim that after a stroke there is greater risk of suffering a recurrent stroke than an MI is not supported by the results (17% n=32 stroke vs 15% n=29 MI).

L338-340

As the reviewer suggests, we have changed the sentence to the following: “However, after having a stroke the chance of having another stroke or having a cardiac event is similar”.

  1. L273-274 Similarly - no statistics are presented to support the statement.

As we have commented in point 5, we have reviewed and modified the sentence.

  1. Much of the discussion merely provides detail of other related studies and reports their outcomes compared to the current study and does not actually discuss or interpret.

L207-233

L220-233

As we have commented in point 1 we have reviewed the entire discussion to correct all the reviewer's comments. Points 3 to 7.

Reviewer 3 Report

The present study is an analysis of the ARTPER cohort, a population study including 520 patients who experienced at least one cardiovascular event. They report a high frequency of further cardiovascular events, usually taking the same form of the first event. The risk of recurrence increased with age, the presence of peripheral arterial disease, antiplatelet therapy, diabetes and an ankle-brachial index <0.9. The analysis is free from major flaws and the results are reasonable, albeit not particularly original. The added value to the established notion that patients with prior cardiovascular events have a higher risk of further events of the same type is very limited.

Author Response

The present study is an analysis of the ARTPER cohort, a population study including 520 patients who experienced at least one cardiovascular event. They report a high frequency of further cardiovascular events, usually taking the same form of the first event. The risk of recurrence increased with age, the presence of peripheral arterial disease, antiplatelet therapy, diabetes and an ankle-brachial index <0.9.

The analysis is free from major flaws and the results are reasonable, albeit not particularly original.

The added value to the established notion that patients with prior cardiovascular events have a higher risk of further events of the same type is very limited.

 Our response to reviewer 1 applies here, as well:

We agree with the reviewer that some of our findings are not new, specially the relationship between the types of first cardiovascular event with the recurrent one. However, ARTPER is an ongoing population-based cohort with more than 10 years of follow-up and almost 4,000 participants enrolled. We are in contact with our participants every 6 months, the accuracy of events are checked by a medical staff, there is a focus on peripheral arterial disease (PAD), and the context is the Mediterranean region*. Several interesting results from our cohort have been published by now, most of them included in high impact systematic reviews or meta-analyses. We think that it is interesting to learn how recurrence works in our cohort and how it relates to PAD to help better identify PAD and its potential effects and provide a better characterization of our cohort. In any case, we have added a sentence recognizing theshortcoming the reviewer points out and we have added some references provided by him/her.

*Please note the French paradox: the paradoxical situation in France with a low death rate from coronary heart disease despite the high consumption of saturated fat in the diet. The French paradox was later extended to other Mediterranean countries such as Spain, Italy, parts of the former Yugoslavia and Greece, where there were differences in both cardiovascular mortality and the incidence of myocardial infarction (MI) with respect to northern European countries. 

Dégano IR, Elosua R, Kaski JC, Fernández-Bergés DJ, Grau M, Marrugat J. Plaque stability and the southern European paradox. Rev Esp Cardiol (Engl Ed). 2013 Jan;66(1):56-62. English, Spanish. doi: 10.1016/j.recesp.2012.07.014. Epub 2012 Oct 15. PMID: 23078876.
